# Experimental Study on Constant Speed Control Technology of Hydraulic Drive Pavers

**Xiujie Yin** [1,*]**, Shujun Yin** [2]**, Hong Zhu** [1] **and Zhihao Zhang** [1]

1    School of Construction Machinery, SHAN DONG JIAO TONG University, Jinan 250357, China
2    Highway School, Chang'an University, Xi'an 710075, China
*    Correspondence: 206047@sdjtu.edu.cn; Tel.: +86-13065091268

**Abstract:** The paver needs superior constant speed performance when paving the pavement. In order to effectively reduce the paver speed fluctuation of the paver, and the wandering deviation from the predetermined track during the paving operation, a control scheme of paver travelling system based on GNSS, Global Navigation Satellite System, is proposed; the scheme can realize open-loop control, closed-loop control, and deviation correction control according to the driver's choice. During closed-loop control, the setting value and the PID controller output of the left wheel are combined to control the speed of the left wheel, as is the closed-loop control of the right wheel. During the deviation correction control, the coordinate provided by the RTK GNSS receiver and the predetermined trajectory line are used to calculate the lateral deviation of the paver. The lateral deviation is input to the right wheel navigation correction PID algorithm. After the calculation, the correction value of the right wheel speed is obtained, which is input to the right wheel PID controller for the deviation correction control. In this paper, the low constant speed performance of the paver, such as during straight driving, turning driving, and driving when resistance changing, was studied by means of experiments. The test results show that when the test paver was running at a speed of more than 2 m/min, the average speed was almost the same. The higher the average speed was, the more stable the speed was. When the paver was less than 1 m/min, its speed fluctuation tended to increase, and its constant speed performance could not be guaranteed. When the test paver hit a movable obstacle at a speed of 5 m/min, which changed the driving resistance, the average speed of the left and right wheels decreased significantly, with a change of about 2.8%, and there was no significant change in the speed fluctuation of the left and right wheels. At the same time, the wandering deviation test proves that the strait-line travelling wandering deviation was basically controlled within 2.5 cm. Without driver intervention, the wandering deviation of the test paver travelling 50 m decreased by about 97.4%, and the constant speed control fluctuation was within 0.2% when the paver travelled at the speed of 5 m/min.

**Keywords:** hydraulic drive; overshoot; constant speed; PID

## 1. Introduction

At present, mobile machines in some industries need to have good low constant speed performance [1] to meet their work requirements, such as the paving of pavers. It is difficult for the mechanical transmission chassis to meet the requirements. Generally, the full hydraulic transmission system with two pumps and two motors is selected. As shown in Figure 1, the chassis drive scheme is composed of two closed hydraulic circuits composed of variable pumps and variable motors. Each hydraulic circuit controls a drive wheel. The hydraulic circuits on both sides can be independently controlled to realize the forward, backward, steering, stepless speed changes and other actions of the paver. Compared with mechanical transmission chassis, pavers with full hydraulic transmission chassis have the characteristics of stepless speed change. Further, constant speed control

is very important, especially since it is still required to achieve stable speed control in low-speed operation.

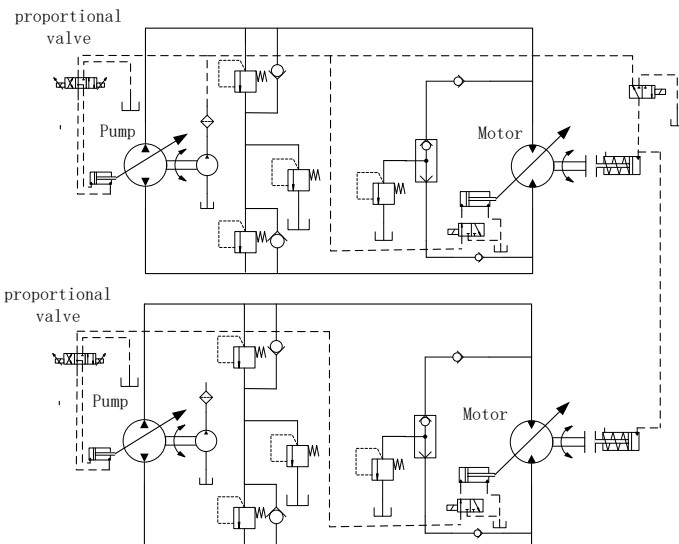

**Figure 1.** Hydraulic transmission schematic diagram of paver.

In 2020, Shi Chendi et al. [2] studied a hydraulic drive spindle speed control based on a fuzzy adaptive PID controller. The fuzzy algorithm was used to detect and adjust the parameters of a traditional PID controller to form a fuzzy adaptive PID controller. It was concluded that the maximum deviation of the designed fuzzy adaptive PID controller was 3.83% less than that of the traditional PID controller, and it could control the speed of the hydraulic drive spindle more accurately. In 2018, Liu Ting et al. [3] designed a PID controller based on the RBF neural network to control the output speed of the pump control motor system. Using Matlab software, the simulation analysis of hydraulic pump control motor system proved that RBF neural network controller had better dynamic characteristics and load disturbance adaptability in controlling motor output speed. In 2021, Huang Yajun et al. [4] proposed a four-pump and four-motor hydraulic drive system and determined the circuit diagram of the hydraulic drive system. Using AMESim software, the simulation model of the unilateral hydraulic drive circuit was established, and the dynamic response of the 565 kW full hydraulic bulldozer was studied. During the starting process of the bulldozer, when the paver speed was 2.6 km/h, the bulldozer still started smoothly. In 2021, Jiang Zhenhong et al. [5] studied the hydrostatic transmission system of a tractor. By using the power bond graph theory, they established a bond graph model of the tractor hydrostatic transmission system. By using SIMULINK simulation software, they simulated and analyzed the dynamic characteristics of the tractor hydrostatic transmission system. They believe that with a certain depth of tillage, the smaller the throttle opening is during normal operation of the tractor, the smaller the fluctuation of fuel consumption rate will be when the soil-specific resistance changes

To sum up, the research on the constant speed control system of hydraulic transmission mostly adopts the simulation method [6–9]. For pavers, the constant speed travelling performance is more important. If the speed was not constant, it would mean that there was an acceleration of the paver, which would affect the force balance of the paver's screed, the screed would be easy to vibrate, and then the flatness of the paved road would be affected. It would also mean that the number of vibrations within the same travel distance of the paver was different, and that the compactness of the paved road would be uneven. A new control scheme of the paver travelling system is adopted in this paper. The scheme can realize open-loop control, closed-loop control, and deviation correction control according to the driver's choice. When the closed-loop control is used, the setting value obtained from control panel and the PID controller output of the left wheel are combined to control

the speed of the left wheel, as is the closed-loop control of the right wheel. The control function of the PID controller accounts for 10%, and the direct control of the speed setting value accounts for 90%. Of course, other combinations can also be used. This closed-loop control scheme can effectively avoid the speed fluctuation caused by PID regulation, while retaining the closed-loop control function of PID. During the deviation correction control, the coordinate provided by the RTK GNSS receiver and the predetermined trajectory line are used to calculate the lateral deviation of the paver. The lateral deviation is input to the right wheel navigation correction PID algorithm and then used to correct the right wheel speed. In addition, this paper will use the experimental method to study the low constant speed performance of a hydraulic transmission paver.

## 2. Design and Discussion of Paver Travelling Speed Control Scheme

### 2.1. Control Principle

The paver travelling control system includes a PLC controller, two speed sensors installed on the output shafts of the hydraulic motors that drive the left and right wheels of the paver, and two electromagnetic proportional valves installed on the variable displacement pumps. The solenoid valves can control the hydraulic oil flow of the variable displacement pumps. The common closed-loop constant speed control is shown in Figure 2a. The speed feedback value of the left wheel is subtracted from the speed set value of the left wheel, and the deviation is input to the PID algorithm module. The PID algorithm is used to calculate and output PWM, which controls the variable pump electric proportional valve, and further controls the speed of the hydraulic motor. The same is true for right wheel control.

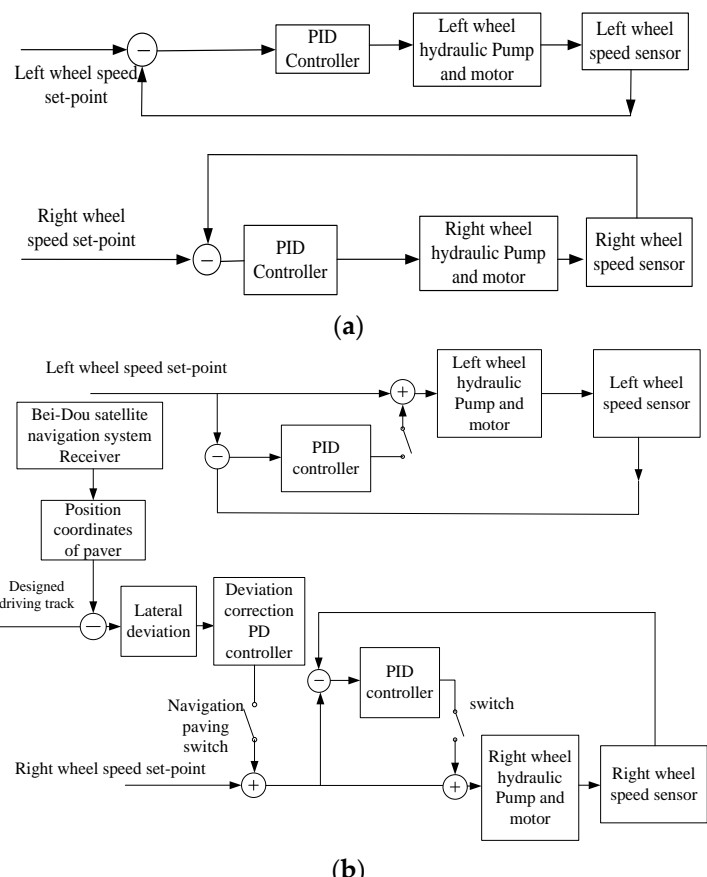

**Figure 2.** Travelling control scheme of paver. (**a**) Current travelling control scheme. (**b**) New control scheme.

In order to effectively reduce the driving wandering deviation of the paver and keep the speed of the paver constant, a new control scheme of the paver travelling system based

on Global Navigation Satellite System is needed. The scheme has speed open-loop control, speed closed-loop control, and deviation correction control. As shown in Figure 2b, a RTK GNSS mobile station is installed on the paver, and a base station is set near the work site. The mobile station is connected with the paver PLC through RS232 to output the actual position coordinates of the paver.

When both the navigation paving switch and the closed-loop switch are in the off position, the closed-loop control and the deviation correction control are not involved in the paver speed control. In this case, the control system calculates the running speed setting values of the left and right wheels according to the speed knob of the paver, and the setting values of the left and right wheels directly control the electromagnetic proportional valves of the variable pumps (Figure 1). When the closed-loop switch is closed, the control system is in the closed-loop state. The output of the PID controller and the speed set value are added together to jointly control the electromagnetic proportional valve of the variable displacement pump (Figure 1), and further control the speed of the hydraulic motor. When the navigation paving switch is closed, the coordinates provided by the RTK mobile station and the predetermined trajectory line are input to the lateral deviation calculation module to obtain the lateral deviation of the paver, which is input to the right wheel navigation correction PID algorithm. Further, the correction value of the right wheel is obtained after the navigation correction PID algorithm operation. The correction value and the right wheel speed setting value are combined to form a new right wheel speed setting value of the paver. The output of the right wheel motor speed sensor is fed back to the input port of the right wheel PID algorithm module, and compared with the new set value of the running speed of the right wheel to form a closed-loop control. When the navigation paving switch is disconnected, the deviation correction control will not participate in the right wheel travelling control. Therefore, the paver travelling control has the following three states:

(1)  Open-loop control: when the paver is in the open-loop state, the left and right wheel speed setting values are calculated according to the speed knob on the panel, and then the left and right wheel speed setting values directly control the displacement pump (Figure 1);

(2)  Closed-loop constant speed control: when the paving operation requires closed-loop control, the set values of the left and right wheels are added to their PID controller outputs to control the variable displacement pump;

(3)  Navigation correction control: when the paver is in the correction control state, the right wheel navigation correction algorithm module performs the calculation according to the principle that the lateral deviation tends to zero, and the algorithm module outputs the correction value of the right wheel, which is combined with the right wheel speed setting value to form a new setting value, so as to realize the deviation correction control of the paver.

To sum up, in this scheme, when travelling at high speed, the paver does not need constant speed control and deviation correction control. In this case, open-loop control is generally selected. When the paver is in usual paving operation, the constant speed closed-loop control is selected. When the paver is in a high-quality operation situation that requires constant speed and deviation correction control, the deviation correction control can be selected. The scheme can easily realize the conversion of open-loop control, closed-loop control, and navigation correction control, which is convenient to improve the operating efficiency of the controller. When the PID controller participates in the speed control of the paver, it means that the PID controller is limited to a small range. It effectively prevents the influence on the travelling speed of the paver when the overshoot of the control system is too large, and effectively ensures the speed control accuracy of the entire control system. When the paver uses navigation correction paving, the accuracy of constant speed control can be further improved according to the actual driving speed of the left and right wheels. In addition, the RTK GNSS mobile station is used to rectify the wandering

deviation of the paver in real time, thus effectively ensuring that the paver travels along the preset travelling track.

### 2.2. Calculation Method of Paver Speed Setting Value

In this scheme, the controller calculates the speed setting values of the left and right wheels according to the speed setting knob A and steering setting knob B on the control panel, as shown in Figure 3. When the paver is paving along a straight line, the steering setting knob B is in the middle position, and the speed settings of the left and right wheels are equal, that is,

$$V_L = V_R = V \tag{1}$$

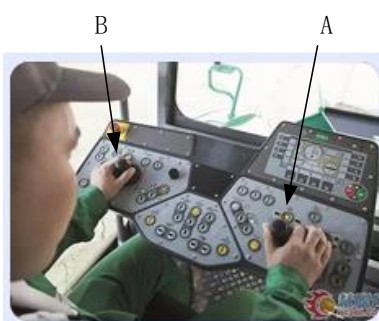

**Figure 3.** Control panel of paver. A—Speed setting knob, and B—Steering setting knob.

According to the forward/backward switch on the control panel, the paver can move forward or backward along a straight line.

When the steering setting knob B deviates to the left or right from the middle position, the paver will be in a turn paving. It is assumed that the running speed of the paver center set by the current speed setting knob A is V.

When the paver turns left,

$$V_L < V_R \tag{2}$$

where

$$V_L = V - \delta V, \tag{3}$$

$$V_R = V + \delta V \tag{4}$$

If the paver turns to the right,

$$V_L = V + \delta V, \tag{5}$$

$$V_R = V - \delta V \tag{6}$$

$$\delta V = \frac{B_X V}{a B_{MAX}} \tag{7}$$

where

V—Paver center speed;
$V_L$—Paver left wheel speed;
$V_R$—Paver right wheel speed;
$\delta V$—Speed difference between wheel and paver center;
$B_X$—Actual value by steering setting knob B;
$B_{MAX}$—Maximum value of steering setting knob B;
a—Paver turning coefficient, a = 2 in this paper.

### 3. Design of New PID Controller

*3.1. Speed Closed-Loop PID Design*

The PID controller can performs proportional, integral and differential operations on deviation e(t).

The expression is:

$$u(t) = K_P \left[ e(t) + \frac{1}{T_i} \int_0^t e(t)dt + T_D \frac{de(t)}{dt} \right] \tag{8}$$

where

$K_P$—Proportional coefficient;
$T_i$—Integral action time;
$T_D$—Differential action time;
Discretize Equation (8),

$$u(k) = K_P \left[ e(k) + \frac{T}{T_i} \sum_{i=0}^{k} e(i) + \frac{T_D}{T} (e(k) - e(k-1)) \right] \tag{9}$$

$$u(k-1) = K_P \left[ e(k-1) + \frac{T}{T_i} \sum_{i=0}^{k-1} e(i) + \frac{T_D}{T} (e(k-1) - e(k-2)) \right] \tag{10}$$

By subtracting Equation (10) from Equation (9), the discrete PID control increment expression can be obtained as:

$$\Delta u(k) = K_P[e(k) - e(k-1)] + K_i e(k) + K_d[e(k) - 2e(k-1) + e(k-2)] \tag{11}$$

where

$K_P$—Proportional coefficient;
$K_i$—Integral coefficient;
$K_d$—Differential coefficient.

According to this paper, the actual assignment range of control board PWM is 0~65,535, and the output of speed sensor is pulse signal of 0~1000 pulse/s. After A/D conversion, the output signal of lift sensor is 0~1024.

The preliminary assignment method of PID controller parameters in this paper is as follows:

$$K_P = \frac{65535}{1024} = 64$$

Accaccording to experience,

$$K_i = \frac{K_d}{4} = 1.6$$

$$K_d = \frac{K_P}{10} = 6.4$$

*3.2. Deviation Correction PD Controller Design*

The differential control is adopted for the deviation correction controller, and the expression is:

$$V(t) = K_{PC} \left[ T_{DC} \frac{del(t)}{dt} \right] \tag{12}$$

where

$K_{PC}$—Proportional coefficient of deviation correction control;
$T_{DC}$—Differential time constant of deviation correction control;
$V(t)$—Right wheel speed correction value of paver;
$el(t)$—Distance difference between left and right wheels of paver.
The expression of $el(t)$ is

$$el(t) = K_{bC}b(t) \tag{13}$$

where

$b(t)$—lateral deviation, the lateral distance between the paver center position and the preset travelling line.

Substitute Equation (13) into Equation (12),

$$V(t) = K_{PC}K_{bC}\left[T_{DC}\frac{db(t)}{dt}\right] \tag{14}$$

Discdiscretize Equation (14),

$$V(k) = K_{bC}K_{PC}\left[\frac{T_{DC}}{T}(b(k) - b(k-1))\right],$$

$$V(k-1) = K_{bC}K_{PC}\left[\frac{T_{DC}}{T}(b(k-1) - b(k-2))\right],$$

The inincremental expression of the deviation correction controller is obtained as follows:

$$\Delta V(k) = K_{DC}[b(k) - 2b(k-1) + b(k-2)] \tag{15}$$

where

$K_{DC}$—differential coefficient, $K_{DC} = K_{bC} \times K_{PC}$.

According to the Chinese National Standard GB/T16277-2021 for pavers, the maximum deviation of the paver 50 m is less than 50 cm without the driver's control. Therefore, the actual value range of the lateral deviation in this paper is 0–50 cm. In order to prevent large fluctuations in the speed of the right wheel from affecting the stability of the paver's speed, the correction value of the paver's right wheel speed is limited to 10% of the actual travelling speed of the right wheel in this paper. According to the maximum paving speed of 24 m/min, the output of the speed sensor is within 0~1024 pulses/s. Therefore, the maximum correction value of the right wheel speed is not more than 100 pulses/s. The preliminary assignment method of the parameters of the deviation correction controller in this paper is as follows:

$$K_{DC} = \frac{100}{50} = 2$$

## 4. Test

### *4.1. Materials and Methods*

#### 4.1.1. Test Site and Equipment

The test site and equipment were as follows:

The runway was dry and hard gravel pavement. The straight distance of the runway was more than 250 m. The curved part of the runway was more than 100 m. The width of the runway was more than twice the width of the test paver. The longitudinal gradient of the runway was not more than 1%, and the transverse gradient of the runway was not more than 1.5%.

The paver RP951 manufactured by Xuzhou Construction Machinery Group in Xuzhou, China was selected as the test object. This paver was driven by a closed hydraulic system with two pumps and two motors. The two electric proportional variable pumps A4VG40EP manufactured by Rexroth in Germany. and the two electric variable displacement motors A6VE80EZ manufactured by Rexroth in Germany. were configured to form two independent closed hydraulic systems.

#### 4.1.2. Test Data Collection Method

In order not to affect the operation of the paver, the left and right wheel motor speed sensor pulse signals are also connected to another EPEC2023 controller manufactured by Epec Oyin Seinajoki, Finland, which is responsible for collecting the speed sensor data of

the hydraulic motor of the paver and uploading the collected speed pulse signals to the notebook through the CAN bus.

### 4.1.3. Constant Speed Control Test

(1) Test of constant speed control when paver travelling straight

The first step was to connect the left wheel and right wheel motor speed sensors of the paver RP951 to the data acquisition system according to the method described in 4.1.2. Then, the second step was to download the software to the original paver controller and power on the data acquisition system to collect data. Further, the travelling speed of the test paver was set at about 5 m/min. Then, the paver was started and driven along a straight line for several minutes. The third step was to replace the controller with the new control scheme and repeat the above steps.

(2) Test of constant speed control when paver turning

The first step was to start the paver. The speed of the paver was set at about 5 m/min. The second step was to quickly turn the steering knob to the maximum position and power on the data acquisition system to collect data. The paver turned automatically and drove for about 50 m. The third step was to replace the controller with a new scheme and repeat the above steps.

Further, the speed was set at about 5 m/min, and then the steering knob was quickly turned to 1/2 of the maximum position, and the above turning test of paver was repeated.

(3) Test of the influence on the low speed stability of the paver when the driving resistance changes

A forklift was placed about 3 m in front of the test paver. The forklift weighed 3 tons and carried 3 tons of iron blocks. The iron blocks were placed on the ground; at the same time, the forklift was braked to increase the friction. The driving resistance of the paver would change when the paver hit the forklift. The second step was to start the test paver and set its speed at about 5 m/min and power on the data acquisition system to collect data. Further, the third step was to replace the controller with a new scheme and repeat the above steps.

(4) Test of low speed stability of paver at different speeds

We kept the paver running straight and powered on the data acquisition system to collect data. In order to test the constant speed performance of the paver with the new control scheme during the driving of the paver, the driving speed of the paver was changed in the order of 2-4-6-5-1 m/min. At the same time, the data of the speed sensors were recorded.

### 4.1.4. Test of Deviation Correction Control

The test distance of paver's straight-line travelling wandering deviation is 50 m according to the Chinese National Standard GB/T16277-2021 for pavers. A starting line, a terminal line, and a lane-marking line were drawn on the test road. The test paver entered from the starting line. The direction of the paver was adjusted to make the longitudinal centerline of the paver coincide with the lane-marking line as much as possible. The paver ran at normal paving speed and passed the test area without driver intervention. A water dripping device was fixed on the paver, and the track formed by water droplets on the road was used as the travelling track line of the paver. The lateral distance between the track line and the lane marking line was measured every 3 m.

Figure 2b and the controller with the original control scheme shown in Figure 2a were tested on XCMG RP951 paver. The lateral distance between the actual travelling line of the paver and the lane marking line was measured and recorded every 3 m; the measured data are shown in Figure 4, where the horizontal ordinate is the distance between the actual position of the paver and the starting line, and the ordinate is the lateral distance between the actual position of the paver and the lane marking line. In order to reduce the error, the

initial track extension line was used as the preset travelling line for the paver travelling straight. The deviation value of the paver is equal to the difference between its actual travelling track line and the preset travelling track line in the same coordinate system.

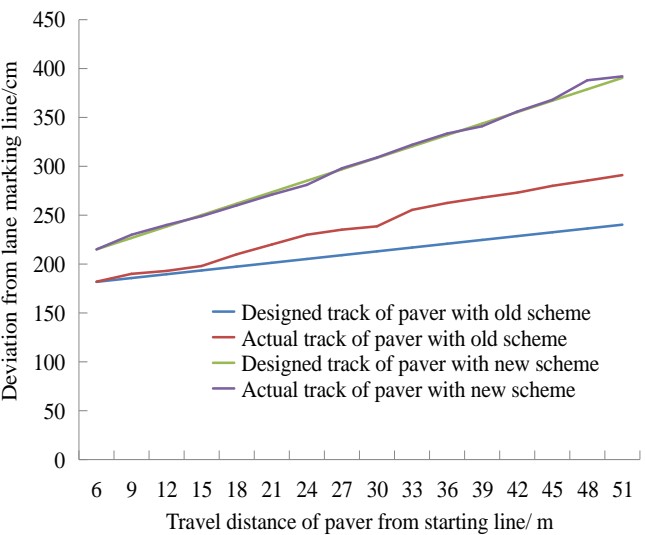

**Figure 4.** Wandering deviation correction test of paver with new scheme and old scheme.

### 4.2. Test Results and Discussion

As shown in Figure 4. The paver straight driving wandering with a new deviation correction control scheme was about 2.4 cm when it travelled 50 m. On the contrary, that of the paver with the original control scheme was about 89 cm. In the original control scheme, the straight driving wandering of the paver occurred in two stages. Firstly, at the beginning, the main factors were the difference in the minimum current of the left and right wheel solenoid valves, the difference in the pump and motor of the left and right wheels under small hydraulic oil flow, the difference in the driving resistance of the left and right wheels, and the difference in the wheel slip rate of the left and right wheels. Secondly, during constant speed driving, the main influencing factors were the different resistance and the different wheel slip rates of the left and right wheels. Especially in the beginning, once deflection is formed, the error will be amplified after travelling a distance. On the contrary, in the new navigation correction control scheme, the influencing factors are included in the deviation correction control where the lateral deviation tends to zero, and are finally eliminated by modifying the speed setting value of the right wheel.

In addition, from the test results, the wandering deviation of the paver with the new navigation correction control scheme was about 2.4 cm, on the contrary, that of the paver with the original control scheme reached about 89 cm when driving straight. The main reason was that when the paver with the new scheme used the navigation correction control, on the one hand, the constant speed control was based on the speed of the left wheel; on the other hand, according to the centimeter level positioning accuracy of the RTK GNSS receiver manufactured by Huace Navigation Co., Ltd. in Shanghai, China, the right wheel speed setting value could be slightly adjusted. This not only ensures the realization of constant speed control and navigation correction control, but also avoids their mutual interference.

The constant speed test results of the paver are shown in Figures 5–8. The ordinate is the speed of the left and right hydraulic motors of the paver. The speed unit is the number of pulses output by the speed sensor per second, and the horizontal coordinate is the paver travelling time.

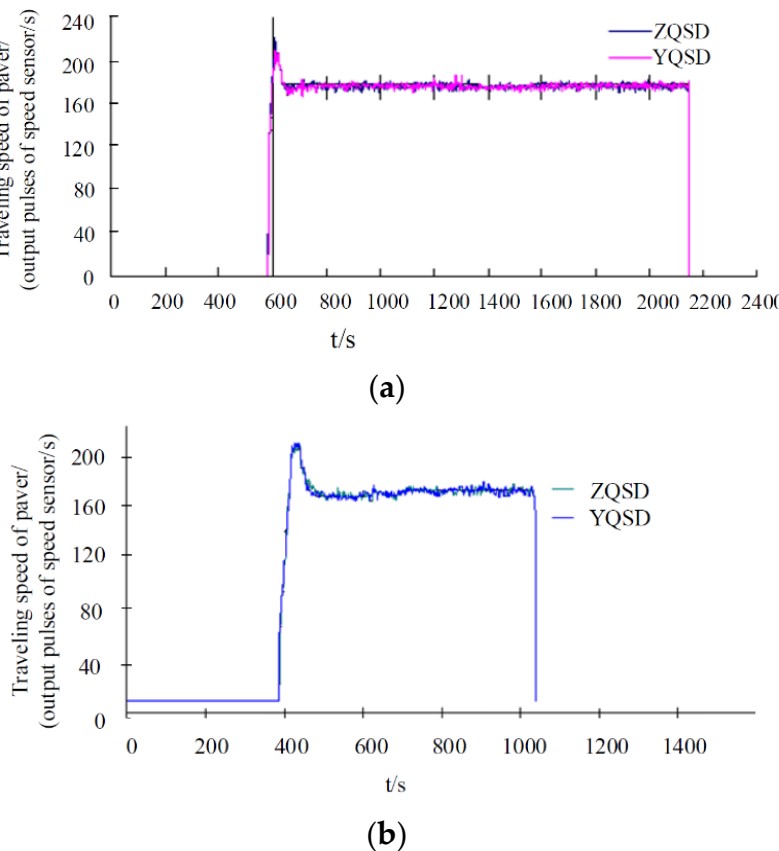

**Figure 5.** Constant speed test of paver when driving straight. (**a**) Constant speed test of the paver with the new control scheme when driving straight. (**b**) Constant speed test of the paver with the old control scheme when driving straight. ZQSD—Speed of the left wheel, and YQSD—speed of the right wheel.

The test results of the paver travelling straight are shown in Figure 5. The test speed of the paver was about 5 m/min, which is the usual paving speed (at this speed, the speed sensor outputs about 180 pulses/s). As shown in Figure 5a, under the control system with the new scheme, the average travelling speed was 180 pulses/s, the whole process was stable without change, and the left and right wheel speeds were almost equal. The control accuracy can be expressed by the speed fluctuation, which was basically kept within ±10 pulses/s. When the step signal is input, the peak speed of the left wheel can reach 238 pulses/s, and the overshoot can reach 32.2%, but the peak speed of the right wheel can reach 210 pulses/s, and the overshoot can reach 16.7%. As shown in Figure 5b, under the control system with the original scheme, the average driving speed was 170 pulses/s, the average value of the whole process changed slowly, the speed of left and right wheels was almost equal, and the speed fluctuation was basically kept within ± 10 pulses/s. When the step signal is input, the peak speed can reach 200 pulses/s, and the overshoot can reach 18%; by comparing the data of the new and old schemes, the new scheme adopted the control method of open loop and PID closed loop in parallel. The PID control function becomes smaller, which can reduce the overshoot of the control system and keep the average speed stable.

The test results of the paver turning are shown in Figure 6. The running speed of the paver center was about 5 m/min. As shown in Figure 6a, under the control of the new scheme, the speed of the left and right wheels changed rapidly after the paver steering knob on the control panel was quickly set to the maximum position, the speed of the left wheel decreased, and the speed of the right wheel increased. The speed curves of the left wheel and the right wheel are almost symmetrical, which is consistent with the differential

Formula (4) and (5). In this process, the right wheel speed had an overshoot and the peak value reached 420 pulses/s. The overshoot was about 16.7%. The average value was stable at 360 pulses/s, which meant that the right wheel of the paver was equal to 9 m/min, and the speed fluctuation was basically within ±10 pulses/s. There was an overshoot at the beginning of the speed change of the left wheel. The minimum speed was close to 0, and the average value was stable at 37 pulses/s (this speed is equivalent to the left wheel speed of 1 m/min). The speed fluctuation could be kept within ±10 pulses/s. However, it also indicates that when the speed was less than 1 m/min, the speed fluctuation had a large proportion in the average speed.

As shown in Figure 6b, under the control of the original scheme, after the paver steering knob on the control panel was also quickly set to the maximum position, the right wheel speed had an overshoot, which was significantly greater than the overshoot shown in Figure 5a; the average speed was about 360 pulses/s, and the speed fluctuation was also within ± 10 pulses/s. The speed curve of the left wheel was basically similar to that of the left wheel in Figure 5a.

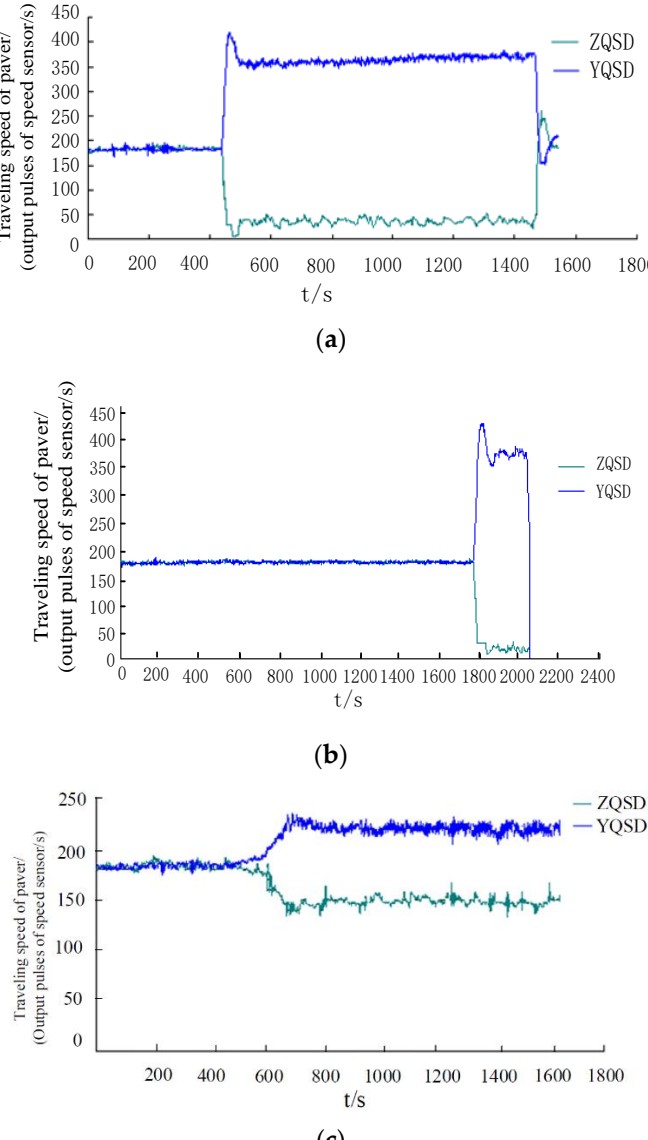

**Figure 6.** *Cont.*

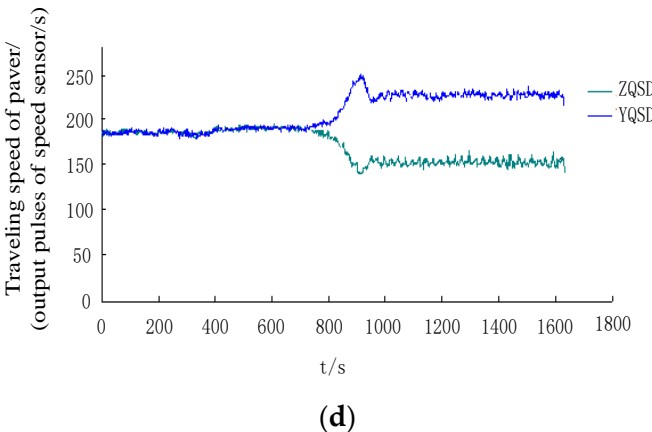

**(d)**

**Figure 6.** Constant speed test of paver when turning. (**a**) Constant speed test of the paver with a new control scheme when turning (maximum position of steering knob). (**b**) Constant speed test of the paver with the old control scheme when turning (maximum position of steering knob). (**c**) Constant speed test of the paver with the new control scheme when turning, (the steering knob is turned to 1/2 of the maximum position). (**d**) Constant speed test of the paver with the old control scheme when turning (the steering knob is turned to 1/2 of the maximum position). ZQSD—Speed of the left wheel, YQSD—speed of the right wheel.

As shown in Figure 6c, under the control of the new scheme, the speed of the left and right wheels changed rapidly after the steering knob was quickly set to 1/2 of the maximum position. Similarly, the speed of the left wheel decreased and the speed of the right wheel increased. Moreover, the speed changing curves of the left and right wheels are also symmetrical, which also conforms to the differential Formulas (4) and (5). At the same time, in Figure 5c, it can be observed that there was no overshoot of the speed of the left and right wheels, and the speed fluctuation was within ± 10 pulses/s. The average speed of the right wheel was stable at 215 pulses/s, which was equivalent to about 6 m/min. The average speed of the left wheel was stable at 145 pulses/s, which was equivalent to about 4 m/min.

As shown in Figure 6d, under the control of the original scheme, the steering knob was quickly set to 1/2 of the maximum position, and the speed of the right wheel had an overshoot. The peak speed reached 250 pulses/s, and the overshoot was about 4.3%. The average speed of the right wheel was stable at 230 pulses/s, which was about 6 m/min, and the speed fluctuation was within ±10 pulses/s. At the same time, after an overshoot of the left wheel speed, the peak value reached 140 pulses/s downward, the overshoot was about 3.7%, and the average value was stable at 150 pulses/s, which was equivalent to about 4 m/min. By comparing the new and old control schemes, under the control of the new scheme, the overshoot was smaller and the speed was more stable when the paver was turning.

Because the dumping truck in front of the paver is often replaced, the driving resistance of the paver will change. The simulation test results, from when the driving resistance of the paver changes, are shown in the Figure 7. The travelling speed was set at about 5 m/min. As shown in Figure 7a, when the paver encountered obstacles, the speed decreased instantaneously by about 10 pulses/s, and then quickly returned to the set value.

As shown in Figure 7b, the paver was in a slight turning state, the left wheel speed was slightly higher than the right wheel speed, and the left and right wheel speed fluctuations were within ±10 pulses/s. It can be seen from the data in Figure 6b that the paver hit an obstacle after driving for about 700 s. At this time, the average speed of the left and right wheels decreased significantly by about 5 pulses/s, and the speed fluctuation of the left and right wheels did not change significantly. The reasons are as follows: Firstly, the new scheme combined the PID closed loop and knob direct control in parallel. Secondly, in the new scheme, the reason why the PID closed loop did not change the output is

that the average speed decreased by 5 pulses/S and was still within the dead zone, so that the control system did not change the output. The original scheme adopted full PID closed-loop control, and the speed was reduced by 10 pulses/s, which exceeded the dead zone of closed-loop control. The PID closed loop worked so that the paver speed could quickly return to the set value.

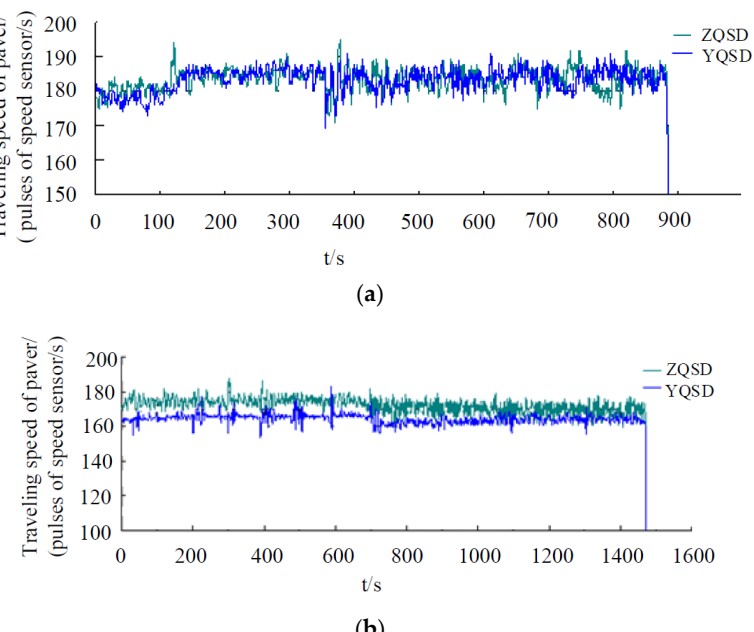

(a)

(b)

**Figure 7.** Constant speed test of paver when driving resistance changes. (**a**) Constant speed test of the paver with a new control scheme when driving resistance changes. (**b**) Constant speed test of the paver with the old control scheme when driving resistance changes. ZQSD—Speed of the left wheel, and YQSD—speed of the right wheel.

The test results, from when the paver was driving straight at different speeds, are shown in Figure 8. When the test paver ran at a speed of more than 2 m/min, which was equivalent to the average of 75 pulses/s, the test results show that the average speeds of the left and right wheel were almost the same, and the speed fluctuation was less than ±10 pulses/s. When the test paver ran at a speed of less than 1 m/min (which was equivalent to the average of 37 pulses/s), the speed fluctuation of the right wheel was significantly greater than that of the left wheel, and the speed fluctuation of the right wheel could reach ±10 pulses/s. The main reason was that there were some manufacturing errors in the hydraulic components of the test paver, which made it impossible for them to be completely consistent. Especially in the case of low speed and small flow, the proportion of hydraulic system leakage gradually increases.

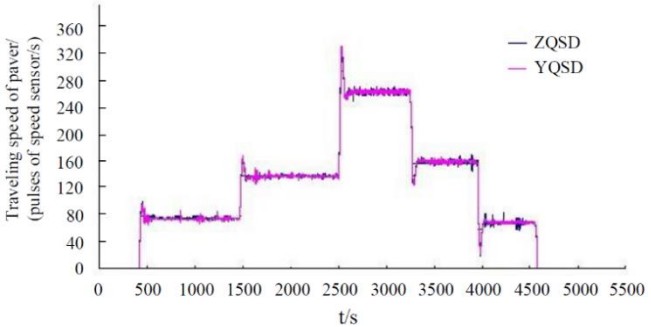

**Figure 8.** Constant speed test of the paver with the new control scheme when speed changes. ZQSD—Speed of the left wheel, and YQSD—speed of the right wheel.

## 5. Conclusions

Aiming at the advanced applications of full hydraulic pavers, such as constant speed control, deviation correction control, automatic driving, etc., a set of paver travelling control schemes based on the Global Navigation Satellite System was provided, which effectively solved the problem that the paver easily deviates from the pre-arranged route. In this scheme, when the paver deviated from the pre-arranged route, the right wheel was slightly adjusted to realize the real-time deviation correction control. At the same time, a travelling control method using open-loop control and closed-loop PID control in parallel was proposed, and the constant speed performance and deviation correction performance of the hydraulic drive paver under the new control scheme were studied by means of experiments. The conclusions are as follows:

(1) When the test paver with the new control scheme ran at a speed of more than 2 m/min, the test results show that the average speed was almost the same, and the speed fluctuation was less than $\pm 10$ pulses/s. The higher the average paver speed was, the smaller the proportion of speed fluctuation was, and the more stable the speed was.

(2) When the test paver was less than 1 m/min, its speed fluctuation tended to increase, and the speed fluctuation reached $\pm 15$ pulses/s. The constant speed of the paver can no longer be guaranteed.

(3) When the paver encountered a movable obstacle, the average speed of the left and right wheels decreased significantly, and the fluctuation of the speed of the left and right wheels did not change significantly. When the running resistance changes, it has a certain impact on the constant speed of the paver, and when the running resistance remains unchanged, it has no impact on the constant speed performance of the test paver.

(4) The travelling system of the paver with satellite navigation could effectively reduce the paver straight-line driving wandering. The wandering deviation of the paver could be controlled within 2.5 cm, and the fluctuation of low constant speed control was less than $\pm 10$ pulses/s. The next work will further study the PID parameter matching of the paver travelling control.

**Author Contributions:** X.Y.: Conceptualization, supervision, funding acquisition, writing—original draft, writing—review and editing. S.Y.: investigation, validation, data curation, writing—original draft, writing—review and editing. H.Z.: investigation, data curation, writing—review and editing. Z.Z.: investigation, data curation, and writing—review and editing. All authors have read and agreed to the published version of the manuscript.

**Funding:** This research received no external funding.

**Data Availability Statement:** The data are not publicly available due to the privacy restrictions.

**Conflicts of Interest:** The authors declare no conflict of interest.

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
