# Peer review of "Experimental Study on Constant Speed Control Technology of Hydraulic Drive Pavers"

_processes, doi:10.3390/pr11020477_

Round 1

Reviewer 1 Report

The experimental results presented adequately demonstrate the benefits of the recommended solutions. The results of this article are also useful for practice. Organized, well-edited work. 

Some notes can be found in the attached file.

It would be worthwhile to test the proposed control with different control parameters. How sensitive is the control to parameter settings?

Reviewer 2 Report

Authors proposed a control scheme of paver travelling system based on Global Navigation Satellite System. The scheme can realize open loop control, closed loop control and deviation correction control according to the driver’s choice. Authors studied the low constant speed performance of the paver by means of experiments.

 Authors did not clearly written what is there contribution in the paper.

 Authors added NAVIGATION CORRECTION CONTROL SCHEME. What is the difference (improvement) in their scheme compared to the schemes of other authors?

 The comments are:

 Some equations do not have numbers (in the line 167, or 171 etc.). Please add the numbers.

 Please explain why in the equation dV=(Bx*V)/(2*Bmax) the 2*Bmax is used, why not only Bmax?

 Line 326: It is written “The controller with the new control scheme and the controller with the original control scheme were tested... “ – It is not clear what is an original control scheme and what is a new control scheme.

 Line 335: There is no introduction of the Figure 4 to 7. Please add short text which will briefly explain what we are going to see in the figure from 4 to 7 and write that the explanation is in the chapter…

 The organization of the paper is confusing. Figures 4 to 7 are in the Chapter 4.1.4, but the explanation is in the chapter 4.2. Figures and descriptions belong in the same chapter.

 At Figure 5 (a), 5 (b) and 5 (d) the mark RQSD is used, which do not appear in the text. What is the meaning of RQSD?

 At Figure 8 it is two times written: “Designed track of paver with old scheme” – one of them is with new scheme

Round 2

Reviewer 2 Report

Authros considered all my comments. Paper can be published in the current form.